# Cell Wall Bulking by Maleic Anhydride for Wood Durability Improvement

**Mingming He [1,†], Dandan Xu [2,3,†], Changgui Li [1,*], Yuzhen Ma [1,†], Xiaohan Dai [2,3], Xiya Pan [2,3], Jilong Fan [2,3], Zaixin He [2,3], Shihan Gui [2,3], Xiaoying Dong [2,3,*] and Yongfeng Li [2,3,*]**

[1] Shandong Academy of Forestry, No. 42 Wenhua Road, Jinan 250014, China; 18615269160@163.com (M.H.); 15053159953@163.com (Y.M.)

[2] State Forestry and Grassland Administration Key Laboratory of Silviculture in Downstream Areas of the Yellow River, Shandong Agricultural University, No.61 Daizong Road, Taian 271018, China; xdandan1995@163.com (D.X.); dxiaohan0315@163.com (X.D.); pxy199612@163.com (X.P.); fjlong@yeah.net (J.F.); hzaixin@126.com (Z.H.); gshihan@126.com (S.G.)

[3] Department of Wood Science and Engineering, Forestry College, Shandong Agricultural University, No.61 Daizong Road, Taian 271018, China

[*] Correspondence: lichg1017@126.com (C.L.); dxiaoying1982@163.com (X.D.); lyf288@hotmail.com (Y.L.); Tel.: +86-531-88557891 (C.L.); +86-538-8240610 (Y.L.)

[†] These authors contributed equally to this work.

**Abstract:** Wood is susceptible to swelling deformation and decay fungi due to moisture adsorption that originates from the dynamic nanopores of the cell wall and the abundant hydroxyl groups in wood components. This study employed as a modifier maleic anhydride (MAn), with the help of acetone as solvent, to diffuse into the wood cell wall, bulk nanopores, and further chemically bond to the hydroxyl groups of wood components, reducing the numbers of free hydroxyl groups and weakening the diffusion of water molecules into the wood cell wall. The derived MAn-bulked wood, compared to the control wood, presented a reduction in water absorptivity (RWA) of ~23% as well as an anti-swelling efficiency (ASE) of ~39% after immersion in water for 228 h, and showed an improvement in decay resistance of 81.42% against white-rot fungus and 69.79% against brown-rot fungus, respectively. The method of combined cell wall bulking and hydroxyl group bonding could effectively improve the dimensional stability and decay resistance with lower doses of modifier, providing a new strategy for wood durability improvement.

**Keywords:** wood; cell wall bulking; maleic anhydride; decay resistance; dimensional stability

## 1. Introduction

As a renewable resource material, wood has been widely used in construction, transportation, and furniture. Its light weight and high strength give wood the potential for use as a renewable advanced material to replace non-renewable structural materials such as steel, alloys, and even carbon nanotubes. This is of great significance for promoting the sustainable development in society [1–5]. However, the drawbacks of wood, such as dimensional instability when exposed to wet conditions and biological degradation when exposed to decay fungi, limit its long-term utilization [6–10]. Although its biodegradable nature is beneficial, from the perspective of sustainable development wood still needs to be modified to obtain better durability when exposed to the above harsh environments [11–15]. The durability of wood with respect to deformation in water and biological degradation due to decay fungi is essentially related to its cell wall components and microstructure [16–20].

Wood cell walls are composed of cellulose, hemicellulose, and lignin, which are rich in hydroxyl groups and can adsorb water molecules through hydrogen bonds; the cell wall possesses a lamellar

microstructure, which carries large amounts of dynamic nano-scale pores and can thus store small polar molecules like water, methanol, and ethanol, etc. [21,22]. When the external environmental water molecule comes across the wood, it firstly enters into the cell lumen with a larger space at the micrometer scale, and then defuses into the dynamic nanopores of the cell wall with the help of the adsorption of hydroxyl groups (i.e., water uptake of wood). This causes the nanopores between the adjacent cell wall layers to expand, manifesting as volume swelling at the macroscale [23]. Meanwhile, water adsorption causes the moisture content inside the cell wall to increase, which is beneficial for the survival of decay fungi due to sufficient moisture supply, thus encouraging the degradation of wood [19,24–26]. Consequently, wood deformation and degradation are closely related to cell wall moisture content; the water uptake of wood originates from the dynamic nanopores and the hydroxyl groups of the cell wall components. Therefore, in theory, preventing water from penetrating into the dynamic nanopores of wood cell wall and especially reaching the hydroxyl groups seems to be a feasible way to improve the wood durability.

Under such consideration, two strategies are widely adopted. One is to block the cell lumen to make it difficult for water to effectively defuse into the cell wall, so as to improve the dimensional stability and decay resistance of wood. The explored modification methods include filling wood cell lumen by physical or chemical means, such as physical filling with poly (glycidyl methacrylate) [27] or phenolic resin [28], physical or chemical filling with polyacrylic resin [29–31], or physical or chemical filling with inorganic silica compounds [32]. However, this strategy implies the use of large amounts of filler, resulting in high cost. Another idea is to fill the dynamic nanopores of the cell wall and/or change the hydrophilic hydroxyl group into a novel hydrophobic group, thereby eradicating the root of the affinity between wood and moisture [33,34]. For example, polyethylene glycol [35], maleic anhydride [36], and silicon compounds [37] have been explored for diffusion into the wood cell wall to fill the dynamic nanopores or bond to the hydroxyl group, preventing water from penetrating into the cell wall and accordingly improving the dimensional stability and even the decay resistance of wood. Certainly, the chemical bonding results in a change of the wood components, and contributes to the improvement of decay resistance [38]. Thus, cell wall treatment is considered as a promising and cost-effective method through drug loading to effectively solve the durability problem. Compared with the previously reported methods which mainly referred to physically filling the dynamic nanopores or chemically bonding to the hydroxyl groups, the strategy proposed in this study is a dual modification method, that is, the use of maleic anhydride (MAn) to improve wood durability (i.e., dimensional stability and decay resistance) via dynamic nanopore filling combined with hydroxyl group elimination. Although acetic anhydride, melamine formaldehyde, sorbitol, and N-methylol melamine have been reported to modify the cell wall for durability improvement in a similar strategy to our work, several drawbacks like the strong acid catalysts employed or formaldehyde release adversely affect their properties and the environment, and thus limit their practical applications [39–43]. Although maleic anhydride was reported to bulk the cell wall and chemically bond to the hydroxyl group, the employed methods still use solvents like DMSO, which are difficult to remove after the reaction, and the vapor phase reaction requires specific equipment [44–46]. This study avoids the above limitations by using acetone as a solvent, with the benefits of easy bulking of the cell wall, easy removal after reaction, and lack of a requirement for specific equipment. The modified wood presents 9% more cell wall bulking with respect to untreated wood, and its dimensional stability and decay resistance are both effectively improved. Such treatment could be inspired to design MAn containing a functional system to bulk cell wall for wood durability improvement.

## 2. Experimental Materials and Methods

### 2.1. Experiment Materials

All chemicals were purchased in China. Chemicals include: maleic anhydride (MAn) (Shanghai Chemical Reagent Factory, Shanghai, China), diacetone acrylamide (DAAM) (Guangzhou Chongshi

Commercial, Guangzhou, China), N,N-dimethylformamide, 1,4-dioxane, dimethyl sulfoxide, and acetone (Tianjin Kermel Chemreagent, Tianjin, China), and methanol, ethanol, and pyrimidine (Hubei Xinjing New Material, Wuhan, China). All the chemicals were directly used without purification. Poplar (*Populus ussuriensis* Kom) is one of the most widely-planted fast-growing trees in China, and its wood is being gradually applied in the industry. It is thus explored in our study. Poplar lumber was obtained from the original plantation areas in Maoershan, located in the northeast of China. Boards measuring 25 mm × 300 mm × 2000 mm (R × T × L) were machined from poplar lumber and were dried at room temperature. Test samples were then cut from these boards. The prepared samples were oven-dried at 103 °C to a constant weight, and they were stored for testing. The oven-dried density of the wood was about 0.33 g/cm³, measured and calculated by the oven-dried weight divided by its volume. All the wood samples were employed in normal state without knot defects and tension/compression features.

The equipment of immersion tank was home-made. Its size was 300 mm × 600 mm (inner diameter × height), and the maximum pressure was 1.7 MPa.

*2.2. Experiment Methods*

2.2.1. Preparation of Modified Wood

MAn was dissolved in acetone to form a mass concentration of 20%, and then a few drops of pyrimidine were added as a catalyst, followed by immersion of the oven-dried wood samples into the above solution at conditions of 0.08 MPa for 20 min, and 0.8 MPa for 20 min in sequence. The pressure herein is the absolute value. After pressure was released, the samples were wrapped in aluminum foil and left at room temperature for 24 h. Finally, the wrapped samples were heated at a certain temperature (80 °C, 90 °C, 100 °C, 110 °C) for certain time periods (2 h, 4 h, 6 h, 8 h) to obtain the MAn-bulked wood (i.e., the MAn modified wood).

To facilitate MAn detection in wood, DAAM (containing nitrogen) and MAN were mixed and dissolved in acetone (20% total concentration) (DAAM/MAN at 1:1 molar ratio). The DAAM was selected for its larger spatial volume, lower solubility parameters as compared to MAn, its reactivity with MAn, and easy solubility in acetone [34].

To effectively swell the cell wall and accordingly improve the bulking effect of cell wall by MAn, seven solvents (methanol, 1,4-dioxane, acetone, N,N-dimethylformamide, dimethyl sulfoxide, pyridine, and ethanol) were employed as solvent of MAn. Each solvent was solely employed to evaluate the swelling efficiency by immersing wood samples in the solvent for 442 h.

2.2.2. Characterization and Properties Evaluation of the Modified Wood

Scanning Electron Microscope (SEM) characterization: A slice sample with size of 0.3 cm × 0.6 cm × 0.3 cm (R × T × L) was cut from the wood sample by a blade, then fixed on the loading platform by adhesive tape, and sprayed by vacuum-gold-sputtering instrument; their morphologies were observed by scanning electron microscopy (FE-SEM, JEM-6610LV, JEOL USA Inc., Peabody, MA, USA) equipped with an energy dispersive X-ray (EDX) detector (GENESIS, EDAX Inc., Mahwah, NJ, USA) for mapping under conditions of high vacuum mode, working voltage of 12.5 kV, and beam spot of 5.0. The recording time for the EDX spectra was 360 s.

FTIR characterization: The wood samples to be tested were first pulverized and sieved through a 100-mesh screen, and then extracted by acetone and tetrahydrofuran in sequence for 24 h. The derived powders of 3~5 mg were employed for the test of the FTIR analysis. The FTIR spectra were obtained using a Nicolet Magna 560 FTIR instrument (Thermo Nicolet Inc., Madison, WI, USA). The sample was placed upon the diamond attenuated total reflection (ATR) accessory of the sample stage and the pressure column was adjusted to the appropriate location for the test. The test parameters were at a resolution of 4 cm$^{-1}$ with a scan number of 32 times.

XRD characterization: The wood samples to be tested were first pulverized and sieved through a 100-mesh screen, and then extracted by acetone and tetrahydrofuran in sequence for 24 h. The crystal structure and crystallinity of the powders were characterized by X-ray diffractometer (XRD, D/max 2200, Rigaku Americas Corporation, Woodlands, TX, USA). The 3~5 mg samples of powders were weighed for characterization. The test parameters included a copper target, ray wavelength of 0.154 nm, scanning angle from 5° to 60°, scanning speed of 4 (°)/min, step of 0.02°, voltage of 40 kV, and current of 30 mA.

Water contact angle measurement: A water droplet of 3 μL was first dropped on the surface of wood sample, and then the dynamic water contact angle was immediately measured by a contact angle goniometer (OCA-15EC, Beijing Eastern-Dataphy Instruments Inc., Beijing, China). The results curve was captured within 120 s by the triangulation method.

Dimensional stability test: The samples were immersed in distilled water for different times, and their corresponding weights and the volumes before and after water uptake were also tested. The dimensional stability was evaluated based on the anti-swelling efficiency (ASE) and reduction in water absorptivity (RWA). The ASE is defined as Formula (1):

$$ASE\ (\%) = 100 \times (VSE_u - VSE_t) / VSE_u \tag{1}$$

where $VSE_t$ and $VSE_u$ are the volume swelling efficiency of treated and untreated wood, respectively. The volume swelling efficiency (VSE) is defined as Formula (2):

$$VSE\ (\%) = 100 \times (V_1 - V_0)/V_0 \tag{2}$$

where $V_1$ is the swollen volume and $V_0$ is the dry volume, respectively [8].

The RWA is calculated through Formula (3):

$$RWA\ (\%) = 100 \times (WAu - WAt) / WAu \tag{3}$$

where $WA_t$ and $WA_u$ are the water absorptivity of treated and untreated wood, respectively.

The WA is defined as Formula (4):

$$WA\ (\%) = 100 \times (W_1 - W_0) / W_0 \tag{4}$$

where $W_0$ and $W_1$ is the sample's weight before and after water uptake, respectively.

The WPG is defined as Formula (5):

$$WPG\ (\%) = 100 \times (W_a - W_b) / W_b \tag{5}$$

where $W_b$ and $W_a$ are the weight of wood sample before and after treatment by MAn, respectively.

The end-matched sample size was 20 mm × 20 mm × 20 mm (R × T × L). Five samples were employed for each evaluation.

Decay resistance test: the experiment was conducted referred to the Chinese forest industry standard-Laboratory methods for the toxicity test of wood preservatives on decay fungi (LY/T 1283-2011). End-matched samples with dimensions of $20 \times 20 \times 10\ mm^3$ (R × T × L) were prepared from each pair of control and the treated wood. Test samples, after being autoclaved at 121 °C for 30 min, were placed on wood feeder chips with dimensions of 22 mm × 22 mm × 2 mm (R × T × L) in the incubator. Each incubator contained three wood samples, and each sample was placed on each wood feeder chip. The relative humidity value inside the incubator was 80%, and the temperature was 28 °C. The test was carried out for 12 weeks, and evaluated by the weight loss rate of the tested samples. A minimum of five specimens were used for the test. The brown-rot fungus was *Gloeophyllum trabeum* (Pers. Ex Fr.) Murr., and the white-rot fungus was *Phanerochaete chrysosporium* Burdsall; these were purchased from China Forestry Culture Collection Center.

The weight loss rate is defined as Formula (6):

$$\text{WLR (\%)} = 100 \times (W_b - W_a) / W_a \qquad (6)$$

where $W_a$ and $W_b$ are the weight of the wood sample before and after fungus attack, respectively.

## 3. Experimental Results and Discussion

MAn is a small-molecule and polar cyclic anhydride compound. In theory, it can diffuse into the dynamic nanopores due to its small volume and high polarity in order to bulk the cell wall, thus occupying the space that the water molecule could originally take up and chemically bonding the hydroxyl group by its active anhydride to prevent the wood cell wall from adsorbing moisture (Figure 1a,b) [34,47]. MAn in one molecule could theoretically bond hydroxyl groups in the weight of two molecules, causing the space between the adjacent molecular chains of the cell wall to become swollen, and the components to be changed from hydrophilic to hydrophobic (Figure 1b,c) [34]. Therefore, this method prevents water molecules from entering into the cell wall by a combination of nanopore filling and hydroxyl group elimination to solve the problem of lower wood durability initiated by moisture.

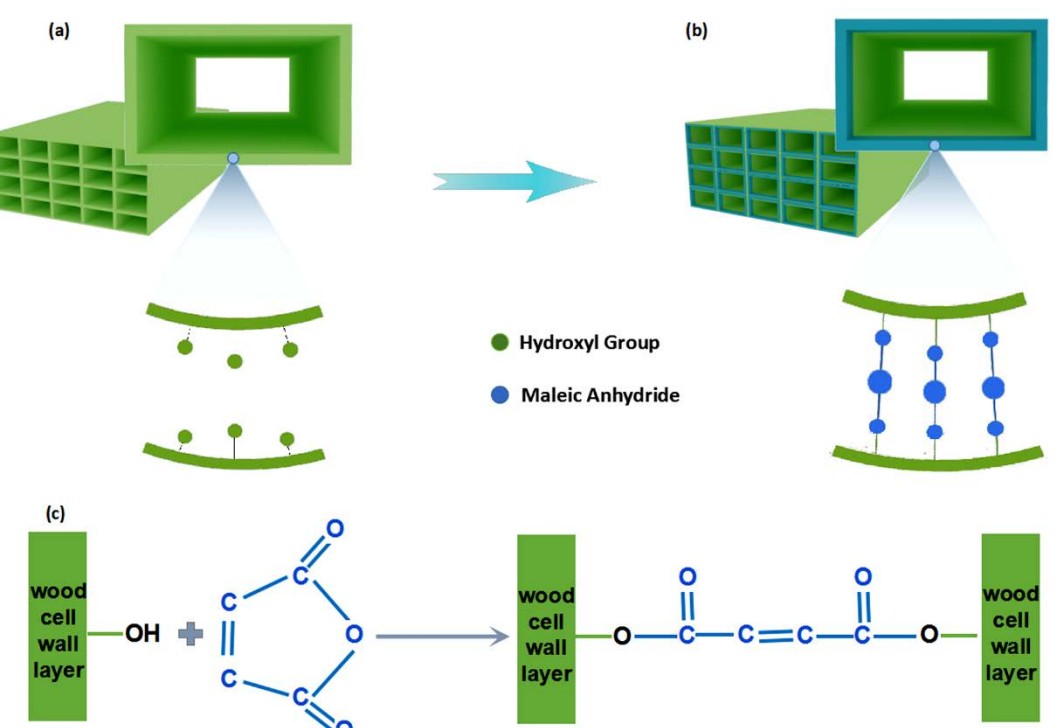

**Figure 1.** Schematic illustration of maleic anhydride (MAn) bulking and reacting with the wood cell wall (**a**) before and (**b**) after MAn bulking. (**c**) The scheme of the expected reaction mechanism of wood cell wall with MAn.

Maleic anhydride as a solid powder compound should be first dissolved into a suitable solvent to form an even solution in a single molecular state, and be then transported by the solvent to deposit in the dynamic nanopores to bulk and bond the cell wall. To characterize whether MAn resides in the cell wall, we explored diacetone acrylamide (DAAM) as an indicator of MAn by dissolving them in acetone together, and used its N element distribution scanned by SEM-EDX to indirectly determine the MAn distribution [34]. After MAn and DAAM diffused into wood cell wall and the chemical reaction, we scanned the cell wall of the modified wood by SEM-EDX. The results show that the N element is well distributed in the cell wall (Figure 2a), and the weight percentage and atomic percentage of N element

accounts for 6.87% and 6.52% of the three total elements (i.e., C, O, and N), respectively (Figure 2b). Such information indicates that DAAM and also MAn penetrated into the cell wall [34]. To effectively swell the cell wall and accordingly improve its bulking effect by MAn, we explored seven solvents to expand the cell wall. The results reveal that except for 1,4-dioxane, all the other six solvents swell the cell wall with volume swelling efficiency (VSE) over 10%, along with a maximum value larger than 20% (Figure 2c). Considering the requirement of easy removal of the solvent after MAn reacting with the components, this study used acetone as the solvent of MAn, as it can effectively swell the cell wall with a VSE of ~11%, and volatilize at a lower temperature without adverse effect on wood.

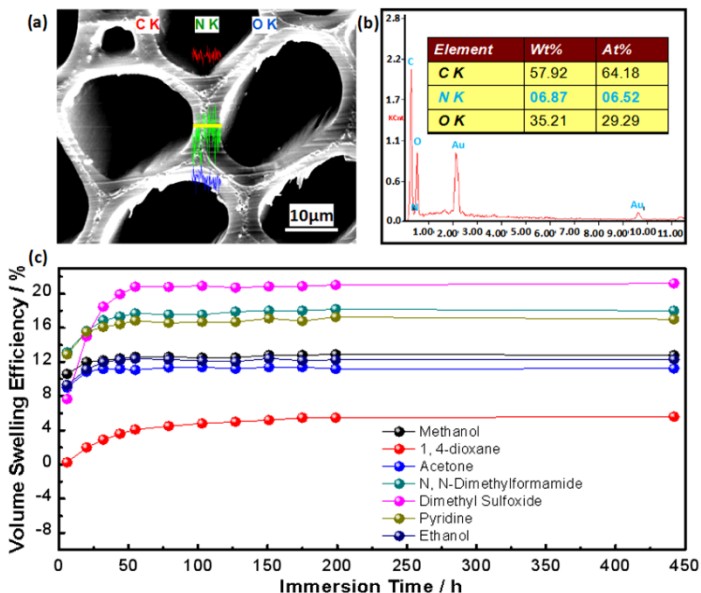

**Figure 2.** Wood cell wall bulking. (**a**) The N element distribution and (**b**) the element content ratio. (**c**) Variations of volume swelling efficiency (VSE) of cell wall swollen by seven polar solvents for immersion time of 442 h.

In order to determine the suitable reacting conditions of MAn and the hydroxyl groups after entering into the cell wall, this study explored the two factors of reacting temperature (80 °C, 90 °C, 100 °C, 110 °C) and reacting time (2 h, 4 h, 6 h, 8 h) in terms of the anti-swelling efficiency (ASE) of the treated wood. Figure 3a shows that when the reacting temperature is 110 °C and the reacting time is 8 h, the maximum ASE of the MAn-bulked wood is 45.85%. At this reacting condition, the weight percent gain (WPG) and VSE of the MAn-bulked wood is ~14.72% and ~8.96%, respectively (Figure 3b).

Under the optimized reaction conditions, the FTIR spectrum (Figure 3c) indicates that the stretching vibration peak of O-H group of the MAn-bulked wood at wavenumber of 3373 cm$^{-1}$ becomes slightly weaker than that of the control wood, and shifts slightly to a relatively lower wavenumber, indicating that the hydroxyl group is partially replaced by MAn and the remaining type of hydroxyl group is changed slightly, which may be caused by the carboxyl group generated from the side reaction of MAn with one single hydroxyl group [34]. The stretching vibration peak of the MAn-bulked wood at the wavenumber of 1733 cm$^{-1}$, representing carbonyl group, is significantly larger than that of the control wood, and similarly, the stretching vibration peak of the C(=O)-O group at the wavenumber of 1240 cm$^{-1}$, the asymmetric stretching vibration peak of the C-O-C group at the wavenumber of 1164 cm$^{-1}$, and the symmetric stretching vibration peak of the C-O group at wavenumber of 1055 cm$^{-1}$ are slightly stronger than for the control wood, respectively. Additionally, a stretching vibration peak, representing the C=C group, appears at the wavenumber of 1635 cm$^{-1}$. All the above information indicate that MAn grafts onto the hydroxyl group of wood [23]. The relative crystallinity of the MAn-bulked wood calculated by the Segal method [30] is 39.94%, which is slightly lower than that of the control wood (42.83%), indicating that MAn chemically reacted with wood components and thus

changed the agglomerate state of wood components (Figure 3d). However, the XRD result indicates that the reaction of MAn and cell wall should be insufficient.

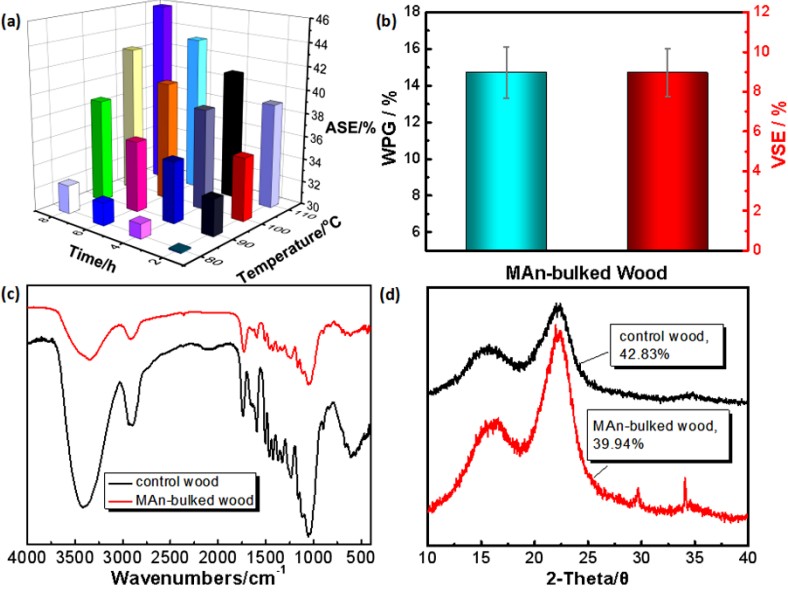

**Figure 3.** MAn bulking and reacting with wood cell wall. (**a**) Variations of the anti-swelling efficiency (ASE) of the MAn-bulked wood under different reacting conditions; (**b**) the weight percent gain (WPG) and the volume swelling efficiency (VSE) of wood bulked by MAn; (**c**) the FTIR spectra of the control wood and MAn-bulked wood; and (**d**) the XRD patterns of the control wood and the MAn-bulked wood.

Both the reduction in water absorptivity (RWA) and anti-swelling efficiency (ASE) of the MAn modified wood present gradually decreased trend within the water immersion time of 228 h, which should be ascribed that water molecule slightly penetrated into cell wall to produce increased swelling; however, the swelling became stable after 24 h of immersion due to the inhibition of cell wall bulking by MAn. Its RWA and ASE reached ~23% and ~39%, respectively, after being immersed in water for 228 h (Figure 4a). Such results indicate good dimensional stability of the modified wood, which should be closely corresponded to the cell wall bulking of Man [47]. Figure 4b further shows that the dynamic water contact angles of the modified wood are much greater than those of the control wood, which could be ascribed to the cell wall bulked by MAn. Figure 4c shows that the weight loss rate (WLR) of the MAn-bulked wood, after decay by the brown-rot fungus, is 23.95%, while that of the control wood reaches 79.28%, representing a 69.79% improvement in the decay resistance of the bulked wood against the brown-rot fungus over the control wood. Similarly, the decay resistance of the MAn-bulked wood (WLR of 5.13%) against the white-rot fungus was increased by 81.42% as compared to that of the control wood (WLR of 27.61%). Both results indicate that the MAn-bulked wood obtains remarkably improved decay resistance. Similarly, compared with several reported wood treatments, this work also presents effective decay resistance of the treated wood (Table 1). The SEM characterization shows that the microstructure of the control wood was obviously degraded by the brown-rot and white-rot fungi, along with collapse of local microstructure (Figure 4d,e,i), while the MAn-bulked wood presents relatively intact microstructure after decay by the both fungi (Figure 4g,h,i). Such information reveals that the decay resistance of the MAn-bulked wood has been improved, indicating the expected purpose was preliminarily achieved. Overall, the improvement of decay resistance of the MAn-bulked wood originally corresponds to the nanopore bulking of the cell wall and the hydroxyl group modification of the component by MAn [48].

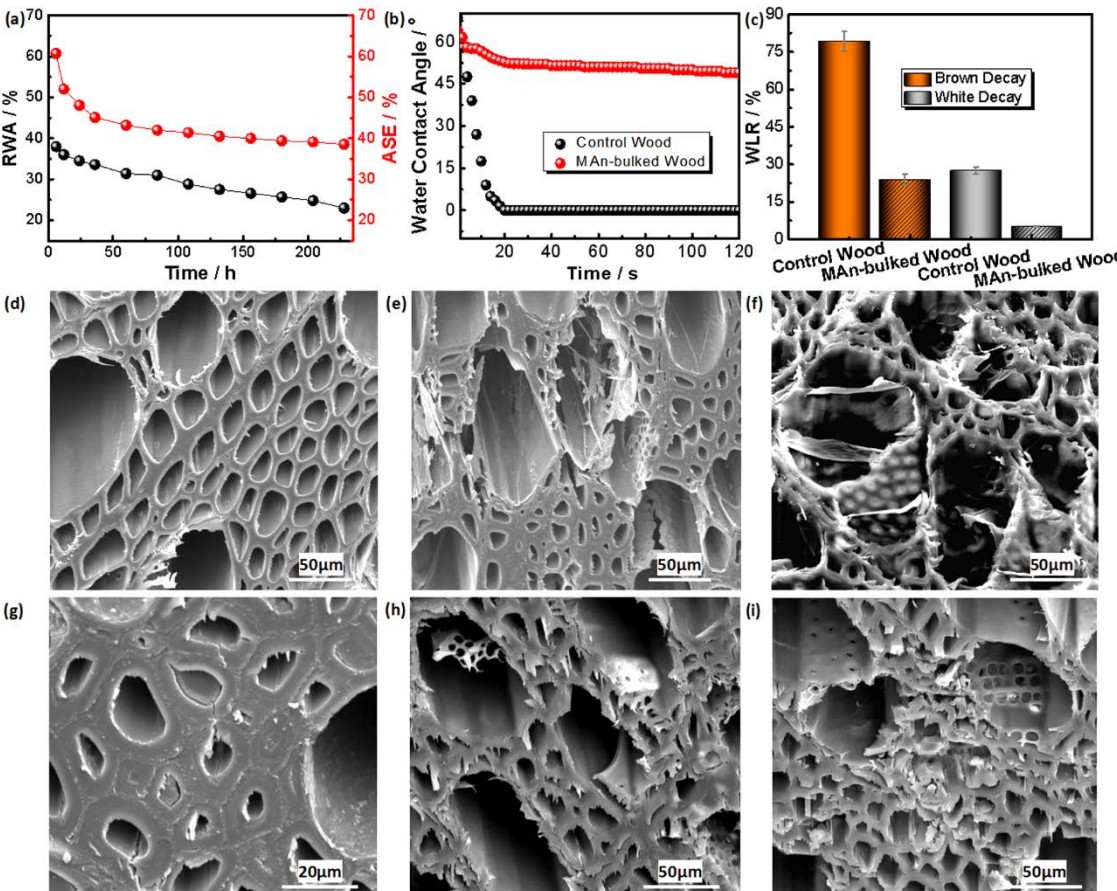

**Figure 4.** Durability comparison of MAn-bulked wood and the control wood. (**a**) Variations of reduction in water absorptivity (RWA) and ASE of the MAn-bulked wood after immersed in water for different times. (**b**) Variations of water contact angles of the MAn-bulked wood and the control wood after water droplet dropped on wood surface for 120 s. (**c**) Comparison of weight loss rates (WLR) of the control wood and the MAn-bulked wood against white-rot and brown-rot fungi. (**d**) SEM morphology of the control wood. (**e**) SEM morphology of the control wood on decay by white-rot fungus. (**f**) SEM morphology of the control wood on decay by brown-rot fungus. (**g**) SEM morphology of the MAn-bulked wood. (**h**) SEM morphology of the MAn-bulked wood on decay by white-rot fungus. (**i**) SEM morphology of the MAn-bulked wood on decay by brown-rot fungus.

**Table 1.** Comparison of decay resistance of our work with other reported treatments.

| Treatment Method | Procedure | Weight Loss Rates (%) | | References |
|---|---|---|---|---|
| | | White-Rot Fungus | Brown-Rot Fungus | |
| Acetylated treatment on wood | Acetic anhydride treatment with weight percent gain of ~10% | ~8 | ~71 | Ref. [49]: Journal of Wood Science, 1999, 45: 69-75. |
| Thermal modification on wood | Thermal treatment: 212 °C for 2 h | 73 | 67 | Ref. [50]: Journal of Wood Science, 2017, 63: 514-522. |
| Polymethylmethacrylate (PMMA) immersion upon wood | PMMA impregnation with weight percent gain of ~59% | ~6 | ~20 | Ref. [51]: International Biodeterioration & Biodegradation, 2011, 65: 1087-1094. |
| Maleic anhydride treatment on wood | Maleic anhydride treatment with weight percent gain of ~9% | 5.13 | 23.95 | Our work |

## 4. Conclusions

(1) Benefitting from the efficient swelling of the cell wall using acetone as a solvent, MAn diffuses into the cell wall and bulks the wood substrates;

(2) The optimum reaction conditions of MAn and hydroxyl group involve a temperature of 110 °C and a reaction time of 8 h; under such conditions, the weight percent gain of the MAn-bulked wood is ~15% and its cell wall bulking reaches ~9%;

(3)The reaction of MAn and hydroxyl groups reduces the water adsorption capacity of the cell wall, causing the RWA and ASE values of the modified wood to increase by 23% and 39%, respectively. Thus, the decay resistance of the modified wood against the white-rot and brown-rot fungus increases by 81.42% and 69.79%, respectively, over the control wood.

**Author Contributions:** M.H., C.L., X.D. (Xiaoying Dong), and Y.L. designed the experiment. M.H., D.X., and Y.M. performed the experiments. D.X. and S.G. drew the figures. X.D. (Xiaohan Dai), X.P., and J.F. carried out the property evaluations. M.H., Z.H., C.L., and Y.L. wrote the paper. All authors commented on the final manuscript.

**Funding:** This research was funded by the Key Special Foundation for the National Key Research and Development Program of China (Grant. No. 2018YFD0600301), Project of Shandong Provincial Agricultural Science and Technology Foundation (Forestry Science and Technology) (Grant. No. 2019LY008); the Key Laboratory of Bio-based Material Science & Technology (Northeast Forestry University), Ministry of Education (Grant. No. SWZ-MS201912); and the National Natural Science Foundation of China (Grant. No. 31700497).

**Conflicts of Interest:** All the authors declare no conflict of interest.

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
