# Peer review of "Cell Wall Bulking by Maleic Anhydride for Wood Durability Improvement"

_forests, doi:10.3390/f11040367_

Round 1

Reviewer 1 Report

This article is a chemical modification study of a poplar wood species by maleic anhydride. The study is rather straight-forward, with an element of uniqueness in the wide range of used scientific instruments. The English language is understandable, but could be improved. 

The introduction describes the various wood modification strategies to reduce the moisture uptake capacity of the wood cell walls, but fail to give a well-balanced state-of-the-art. Firstly, the use of maleic anhydride as a modification agent is not properly reviewed, lacking the contributions of Matsuda (1993), Rozman (1997), Roussel (2001),.... The idea of combined bulking and hydroxyl group substitution is also not new, with e.g. acetic anhydride modification. Nor is the idea of combined bulking, hydroxyl group substitution and crosslinking, with e.g. formaldehyde modification and many other examples. 

I have a few specific questions regarding your work:

  • Why do you consider DAAM a good indicator for MAn diffusion in the wood cell wall? What made you decide to use this particular molecule? Does it have similar bonding properties as MAn?
  • How do you compare the FTIR spectra of control and MAn-bulked wood? Directly, or after vertical shifting and scaling the spectra? - When I shift/scale the control wood spectrum onto the MAn-bulked wood spectrum, there is hardly any difference in the OH-band, only the carboxyl group bands are notably different. Are you sure that reaction has taken place, or is there just physical absorption of MAn?
  • What is happening during the water immersion test that explains the loss of ASE (Fig 4a)?
  • Why has Fig. 4g a larger magnification than the other SEM-pictures? It looks like, the fungally degraded wood (4h and 4i) has lost cell wall volume -- did the MAn leach out? 
  • Did you confirm the bonding reaction of MAn with the wood cell wall by measuring WPG after Soxhlet extraction with acetone? 

The reviewer is altogether not convinced that sufficient MAn bonding has taken place.

Please, avoid the use of "the biodegradation has improved by X %", be more precise to state that "the weight loss in a 12-week fungus test has dropped by X%".

Finally, I don't think MPDI allows to use abbreviations in the reference list of both author names (first and family names), such as "J., W.". Note, also that the non-Asian names must be specified otherwise, e.g. Holger Militz as H. Militz, Emil Thybring as E. Thybring (instead of Holger M. and Emil T.). 

Reviewer 2 Report

The work concerns an important issue which is the resistance of wood to changing environmental conditions. However, it requires some supplement information.
1. It would be good if the authors justified the choice of maleic anhydride for this purpose. Until now, acetic anhydride was used to increase the stability of the wood, so it would be good to refer to these works.
2. What is the reason of soaking the samples in water 228 h?

Reviewer 3 Report

Dear authors,

The article "Cell Wall Bulking by Maleic Anhydride for Wood Durability Improvement" does not show orginality. The paper is like a review showing that wood can be modified with maleic anhydride to improve physical properties (dimensional stability, reduction in water absorptivity, biodegradability...), however, it presents only their work. So if it is an article, they need, a novelty in this study. What did you do that was not done in the scientific world?

Moreover, the major problem in this paper is that we don't know what material was modified. What was the wood species (modification of wood depends on the wood species) How was the wood at the beggining (moisture content, dimensions, where was found the wood...) without this information it is not possible to review this article.

The article is not well written because there are missing elements in the materials and method such as the wood preparation, wood modification-post treatment.... However, there are some details of materials and method in the results and discussion (preparation of maleic anhydride solution, temperature, time of reaction); which is not the place to talk about. Moreover, why the modified wood before FT-IR or SEM is extracted with acetone? it is not extracted before the dimensional stability test or the fungi resistance, maybe it changes the result when extraction is made.

Details of this study are fuzzy. Indeed we don't know the wood species but we don't know either the white and brown fungis used for the experiments. An analysis in figure 2 shows the bulking efficiency of the wood with different solvent. Why is this important? where was it in the material and method?

The conclusion can't be understood because there are missing elements in the article. 

Round 2

Reviewer 1 Report

The authors gave full consideration of the reviewer's comments.